# Long-term effects of buried vertebrate carcasses on soil biogeochemistry in the Northern Great Plains

**Sarah W. Keenan** [1]*, **Scott R. Beeler** [2]

**1** Department of Geology and Geological Engineering, South Dakota School of Mines and Technology, Rapid City, South Dakota, United States of America, **2** Engineering and Mining Experiment Station, South Dakota School of Mines and Technology, Rapid City, South Dakota, United States of America

* sarah.keenan@sdsmt.edu

**Data Availability Statement:** All relevant data are within the paper and its Supporting Information files.

**Funding:** Funding was provided by the South Dakota School of Mines, Department of Geology

## Abstract

Decomposing vertebrates impact ecosystems by stimulating animal, insect, and microbial scavengers, perturbing biogeochemical cycles, and transferring elements back to the environment. Most studies exploring the impacts of vertebrate decomposition focus on surface decay scenarios over timescales of days to years. Accordingly, our knowledge of ecosystem impacts of vertebrate decomposition in burial contexts and over longer time scales is limited. In 2000, six animal carcasses were buried in a shallow grave (<1.0 m) and allowed to decompose naturally until partial excavation in 2021, enabling evaluation of long-term soil biogeochemical responses to decomposing vertebrates. Soils were sampled along three vertical transects from the surface to the bone-bearing layer (~40 cm depth) and below. Comparison of the physical and chemical properties of the grave and control soils from equivalent depths indicate significant perturbations even 21 years after burial. Notably, soil pH was significantly more acidic in grave soils (p = 0.0296), and conductivity was significantly elevated (p = 0.0009). Grave soils were significantly enriched with respect to nitrogen stable isotopes, exhibiting $\delta^{15}$N values of 10.48 ± 3.6‰, which is ~5‰ greater than controls. Carbon and nitrogen content was also disrupted in the burial, with five times more nitrogen in the bone-bearing layer and almost double the carbon. Water and acid-based extractions of soils revealed significant differences between grave and control soils, driven largely by calcium, phosphorus (P), magnesium, and iron concentrations. P concentrations in acid extracts were significantly enriched at the bone-bearing layer, suggesting release of P from the bones. This study demonstrates that decomposition may result in long-lived impacts to burial environments and soil biogeochemistry, even after soft tissues decay. While not typically considered in ecosystem models, buried remains contribute to soils for decades or longer, and soil biogeochemistry serves a critical role in facilitating or preventing the long-term preservation of bone.

## Introduction

The decomposition of vertebrates provides a critical mechanism for returning elements such as carbon (C), nitrogen (N), phosphorus (P), and calcium (Ca) back to soil ecosystems [1].

and Geological Engineering and the Office of Research Affairs to SWK. The funders had no role in study design, data collection and analysis, decision to publish, or preparation of the manuscript.

**Competing interests:** The authors have declared that no competing interests exist.

Decaying animals result in the development of biogeochemical hotspots, stimulating micro- and macro-fauna on a variety of spatial and temporal scales ranging from days (i.e., [2]) to years (i.e., [3]). Animal decomposition can result in long-lived, multi-year impacts on environmental systems including soil [4], water [5], and vegetation [6]. Particularly in areas with nutrient limitation, such as prairie ecosystems, the impacts of animal decomposition can span years to decades [1, 4, 7].

The overwhelming majority of studies characterizing the impacts of animal decomposition hotspots on soil biogeochemistry focus on surface decay settings (Table 1) [8]. Logistically, it is easier to observe, sample, and monitor surface decay experiments compared to subsurface or burial systems. Surface decay experiments also lend themselves more readily to repeated

**Table 1. Synthesis of selected studies and key results focusing on biogeochemical changes to soil during vertebrate decomposition in surface and subsurface (burial) settings.** Temporal duration is presented as Accumulated Degree Days (ADD), when included in publications, and calendar days. N =? indicates that the number of individuals is unknown or was not described.

| | Decay System (n = number of individuals) | Region or Environment | Key Results | Temporal Duration (Accumulated Degree Days [ADD] and Calendar Days) | Reference |
|---|---|---|---|---|---|
| Surface | Humans (n = 5) and pigs (n = 5) | Temperate forest, Knoxville, Tennessee, USA | Enhanced microbial ammonification and nitrification; humans and pigs differ in biogeochemical impacts to soils | 815–1222 ADD; 35–144 days | DeBruyn et al. (2021) [9] |
| | Rabbits (n = 25) | Grassland, Canberra, Australia | No significant change to C; increased pH, conductivity, $NH_4^+$; gradual and persistent increase in P throughout experiment | 20 days | Quaggiotto et al. (2019) [2] |
| | Beavers (n = 5) | Temperate forest, Oak Ridge, Tennessee, USA | Ammonium enrichment early, followed by enhanced nitrification; elevated N stable isotopes; elevated pH, conductivity | 2591.7 ADD; 135 days | Keenan et al. (2018a) [10] |
| | Kangaroos (n = 18) | Temperate woodland, Canberra, Australia | Lasting input of amino acids and proteins; significant impact on microbially-mediated N cycling | 24 weeks; approx. 168 days | Macdonald et al. (2014) [11] |
| | Salmon (n = 12) | Semi-arid riparian forest, Moscow, Idaho, USA | Soil dissolved organic carbon and dissolved total nitrogen significantly increased | 300 ADD; 16 days | Wheeler et al. (2014) [12] |
| | Humans (n = 2) | Southeast Texas, USA | Elevated pH, conductivity; depleted sulfate; movement of decay products downslope | 5779–5469 ADD; 288–248 days | Aitkenhead-Peterson et al. (2012) [13] |
| | White-tailed deer (n = 40) | Northern hardwood forest, Michigan, USA | Reduction of herbaceous cover; increased phosphorus, ammonium, potassium, and nitrate | 15 months; approx. 420 days | Bump et al. (2009a) [14] |
| | European bison (n = 6) | Białowieza Primeval Forest, temperate, Poland | Elevated pH up to 6 years postmortem; heterogeneous soil composition after 7 years (variable calcium, nitrate, etc.) | 1 to 7 years; approx. 2550 days | Melis et al. (2007) [15] |
| Subsurface | Humans (n = 3) | Temperate forest, Knoxville, Tennessee, USA | Nitrogen pools (ammonium, dissolved organic nitrogen) elevated; microbial enzymatic activity elevated | 4 years; approx. 1460 days | Keenan et al. (2018b) [16] |
| | Pigs (n = 12) | Ottawa, Canada | Graves exhibit higher flux of $N_2O$ and $CO_2$ compared to controls; emissions greater in shallower graves | Sampled over 3 years; approx. 1200 days | Dalva et al. (2015) [17] |
| | Pigs (n = 32 limbs) | Ankara, Turkey | Soil type impacts decay rates and soil biogeochemistry; pH decrease in organic and loamy soils; elevated $CO_2$ compared to controls at 3 months in all soils | 6 months; approx. 180 days | Tumer et al. (2013) [18] |
| | Elephant (n = 1); Zebra (n of at least 6); Buffalo (n =?); Watsui (n =?) | Parc Safari Africain, Hemmingford, Quebec Canada | $CH_4$ low within soil pores, elevated above graves; $CH_4$ elevated in graves undergoing active decay | 15 years or less (multiple graves) | Dalva et al. (2012) [19] |
| | Humans (n = 6) | Temperate forest, Knoxville, Tennessee, USA | Variable pH response, with no change up to 2.1 unit increase | 1 to 12 months; approx. 30 to 365 days | Rodriguez and Bass (1985) [20] |

sampling without significantly disturbing the decaying carcass or carcasses. In contrast, each sampling time for a subsurface experiment will likely perturb the experimental system, particularly if the site must be partially or completely excavated to reach the areas of interest. This often results in single sampling timepoints after a period of burial and decay, typically on timescales related to shorter term projects (i.e., thesis or dissertation research, grant-related activity).

Due to these limitations, subsurface decomposition settings are still largely understudied, which greatly inhibits direct comparisons between the surface and subsurface. Studies of decay in the subsurface, particularly those that examine effects on soil biogeochemistry and longer-term studies (i.e., beyond several years), represent significant knowledge gaps for ecology. Understanding subsurface decomposition is also of key importance to the interpretation of animal remains in archaeological and paleontological contexts, which are largely composed of materials preserved in burial environments [21]. Additionally, this knowledge will have applications in agricultural practices (e.g., disposal of animal remains; [22]) and forensic sciences (e.g., identification of clandestine graves; [23, 24]). Therefore, investigating subsurface decomposition and its similarities and differences to surface decomposition is of key importance to a variety of disciplines.

Surface and subsurface systems differ significantly with respect to the physical, chemical and biological processes expected to impact decomposition rates (Fig 1). Surface systems are relatively "open systems" with respect to the input and output of fluids, gases, and biology. Surface decay experiments are more directly influenced by climate (precipitation, temperature, freeze/thaw), and insect and vertebrate scavengers compared to burials. Accordingly, surface systems typically experience more pronounced and rapid changes to moisture availability, UV radiation, gas diffusion and exchange, and physical processes on diurnal and seasonal time-scales [2, 25]. The role of photosynthesis and vegetation in inorganic carbon fixation and nutrient uptake also results in more spatially and temporally dynamic processes in surface systems. As a consequence of these physical, chemical, and biological differences, rates of decomposition are typically greater in surface systems compared to burials [8, 20, 26].

Conversely, subsurface systems, or burials, represent a relatively "closed system" with restricted input or output of fluids, gases, or microbial and invertebrate fauna and decay processes are often diffusion limited [8, 10, 18, 26–29] (Fig 1). In Keenan et al. [10], inhibition of nitrification potential, reduced nematode abundances, elevated respiration rates ($CO_2$ production), the presence of carcass-derived organic C and N compounds, and adipocere all suggested that the subsurface decay system was largely isolated from surrounding soils. Reduced enzymatic activity potential rates (leucine aminopeptidase or LAP and phosphodiesterase or PDE) were also suggestive of oxygen limitation controlling microbial activity despite the presence and availability of C and N resources [10]. Similar observations in other decomposition studies (e.g., [30]) suggest that burials are more likely to become anaerobic and to persist under anaerobic conditions for prolonged periods of time. These anaerobic conditions favor the growth of microorganisms and invertebrates that tolerate or thrive under oxygen-limited conditions [18]. In terms of soil biogeochemistry, prolonged anaerobic or hypoxic conditions results in significantly lower rates of soft tissue breakdown, which likely prolongs the persistence of carcass-derived compounds in the soil. However, the extent and the longevity of decay products in burial settings is not well established. Oxygen limitation also favors the formation of adipocere. Adipocere development typically slows decomposition, and grave soils with adipocere formation are observed to contain lower microbial biomass and microbial activity despite the presence of abundant C and N pools [26].

Soil type, which is in part controlled by climate and regolith (or rock type), may be more important for controlling decomposition processes in subsurface systems compared to the

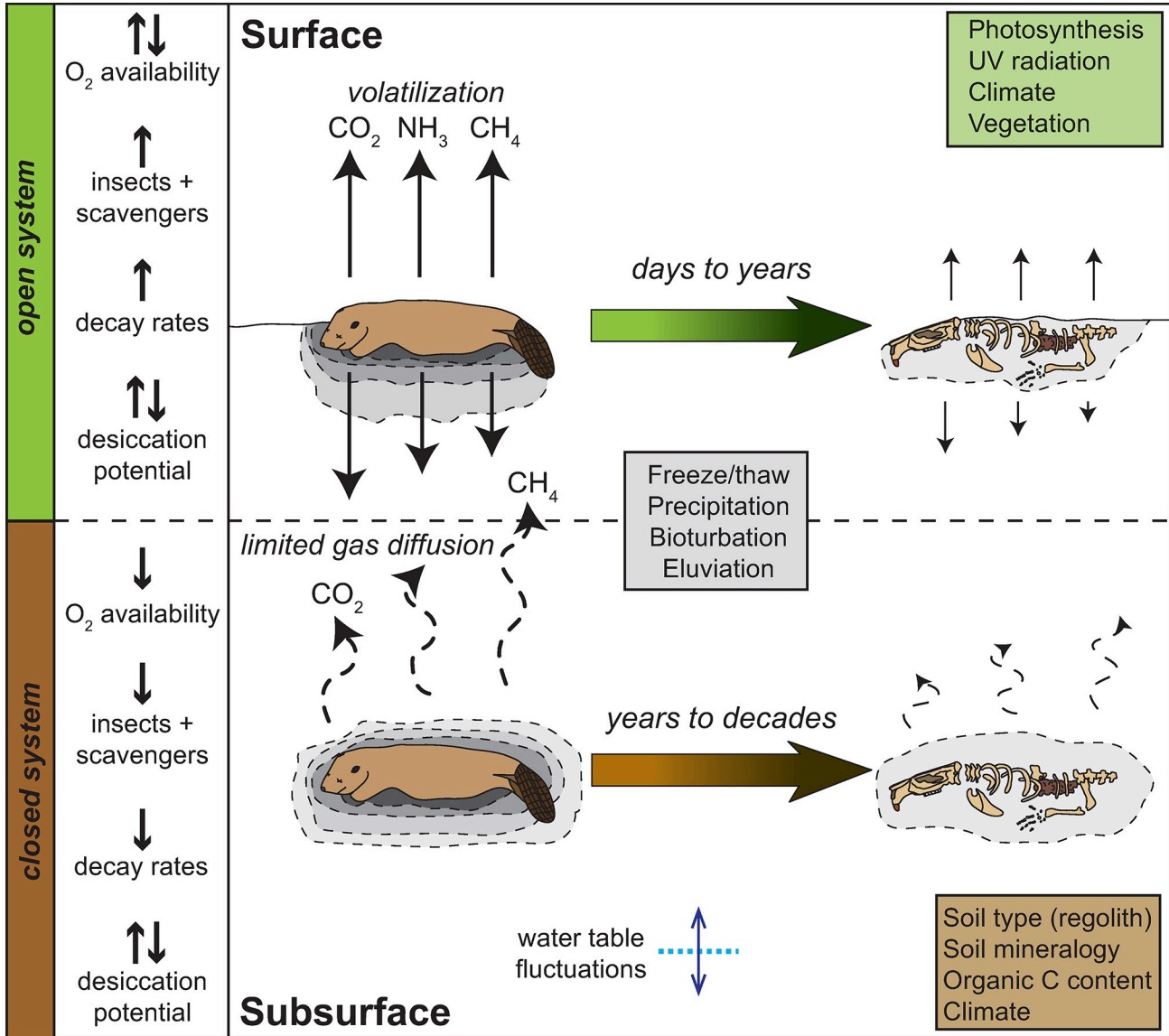

**Fig 1. Schematic diagram comparing surface and subsurface vertebrate decomposition processes.** Surface decay occurs in an open system with respect to gas and fluid exchange. In contrast, subsurface or burial decay occurs in a closed system, where gas diffusion limitations exist. Notable differences between oxygen availability, and the amount and intensity of insects and other scavengers results in lower decay rates in subsurface settings. Desiccation potential can vary depending on soil type, climate, and precipitation.

surface. Soil with greater porosity and lower organic carbon content may facilitate greater rates of moisture loss, which could promote desiccation, slowing decomposition rates dramatically (i.e., thousands of years; [31]). Burial systems are also greatly influenced by the water table level. Water infilling of graves or a rise in the water table can result in reducing conditions and can also favor adipocere development [26]. Burial depth as well as soil temperature are also important for dictating decay rates and timing [30].

At present, the overwhelming majority of decomposition studies focused on subsurface decomposition relate to human interments, particularly understanding how long it takes bodies to decompose (or what remains after specified timeframes; e.g., [20]) and the potential impacts to groundwater, surface water, or soil (e.g., [32]). The focus in burial decomposition studies is typically placed on understanding soft tissue decay, which is often critical in forensics

(forensic taphonomy) for preserving vital evidence, for the visual identification of remains, and for determinations of postmortem interval [24]. While it is widely thought that soft tissue decay is complete within 3–12 years in burial settings [26], the presence of bones and teeth after vertebrates are skeletonized is likely to impact soils long after soft tissues are removed. Understanding the longer-term (bio)geochemical effects of these hard tissues may have implications for ecology, geology, archaeology, and forensics.

As a step towards understanding longer-term effects on soil biogeochemistry in subsurface decay settings, soil from a multi-individual grave containing six vertebrate carcasses (five North American beavers and one porcupine) was examined after 21 years of burial. The burial site is in a prairie in western South Dakota near Hermosa and represents one of the longest modern (i.e., not archaeological) subsurface decomposition systems studied to date, capturing a critical interval after soft tissue degradation. Geochemical characteristics ($\delta^{15}$N, $\delta^{13}$C, C/N, major and trace metals) of burial soils were compared to adjacent, non-burial control soils to assess the decadal scale effects of animal decomposition on soils in a burial setting. Based on prior studies of subsurface decomposition on shorter timescales, soils associated with the multi-individual grave were expected to be significantly altered and retain geochemical signatures of decomposition. In particular, the site was expected to exhibit nitrogen stable isotopic enrichment, demonstrating protracted disruption to nitrogen cycling, and low carbon to nitrogen ratios, reflecting perturbations to carbon pools. The results of this study provide important insights into the temporal longevity of soil biogeochemical modification following vertebrate decomposition and add to our understanding of the impacts of carrion on terrestrial ecosystems.

## Materials and methods

### Site description and location

In fall of 2000, five North American beavers (*Castor canadensis*) and one porcupine *(Erethizon dorsatum)* were placed in a shallow grave (<0.2 m), and covered with soil excavated from the surrounding area with a backhoe (total burial depth ~ 1.0 m). The animals were originally buried with the intent of using the site for future teaching and/or recovery of bones for the South Dakota School of Mines and Technology Museum of Geology recent vertebrate collections. Animals were salvaged from local wildlife officials and did not require special permits prior to burial. Burial occurred on private land with landowner permission. The site was left undisturbed after animal burial for 21 years, unintentionally providing an opportunity to examine the long-term impacts of decay in a subsurface setting. Partial mound excavation and sampling occurred in November 2021 with landowner permission (S1 File).

The burial site is located northeast of Hermosa, South Dakota, USA (Custer County), adjacent to Spring Creek (Fig 2). The soil type is loam to very fine sandy loam. The protolith is the Cretaceous-aged Pierre Shale Formation as well as fluvial overbank deposits (silt, clays, and sand) from the adjacent creek (Spring Creek). Adjoining land to the south and west was used for cattle ranching until 2013. No cattle or other animals routinely grazed near the burial site. The site stands out as a slight topographic high (<1.0 m total height) within a flat, shortgrass prairie. The mound was completely re-vegetated at the time of sampling. Some areas on the top of the ellipse-shaped mound were disturbed by small burrows, which extended up to 0.25 m into the subsurface. Burrows did not reach the burial layer based on visual observations of the mound and at the locations sampled for the present study.

### Sample collection

In order to locate the extent of the buried beavers, a narrow (~0.4 m-wide) trench was excavated through the northern margin of the mound to a depth of 2 m. The trench continued

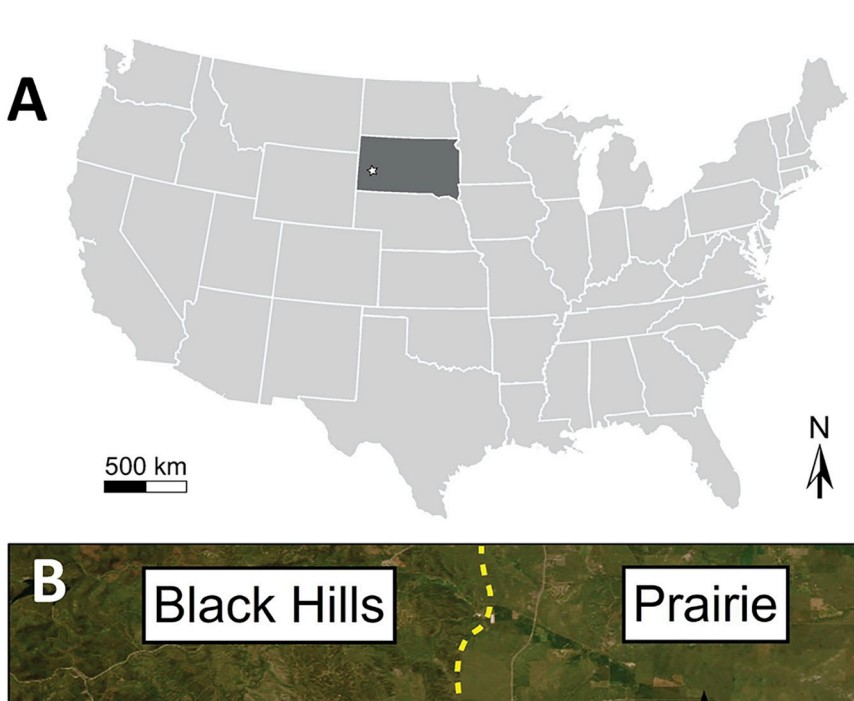

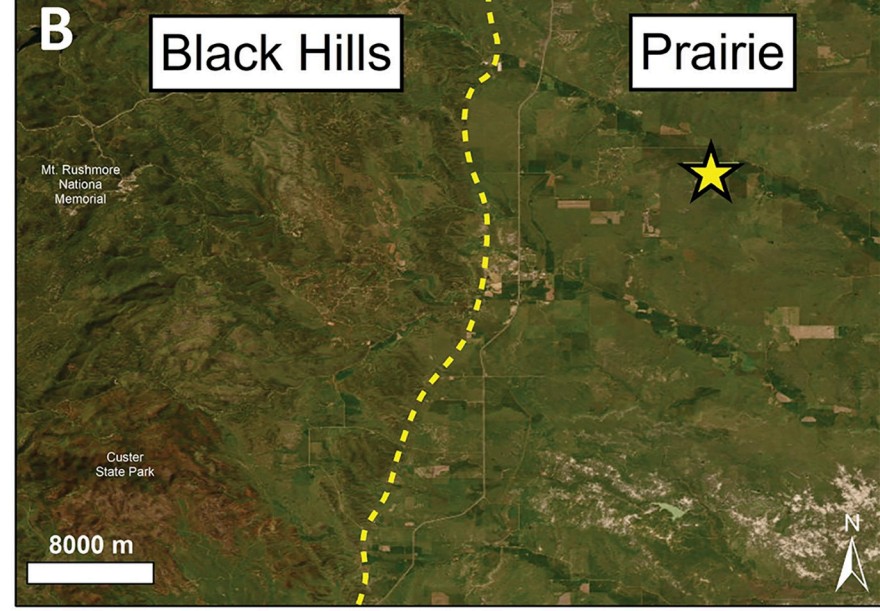

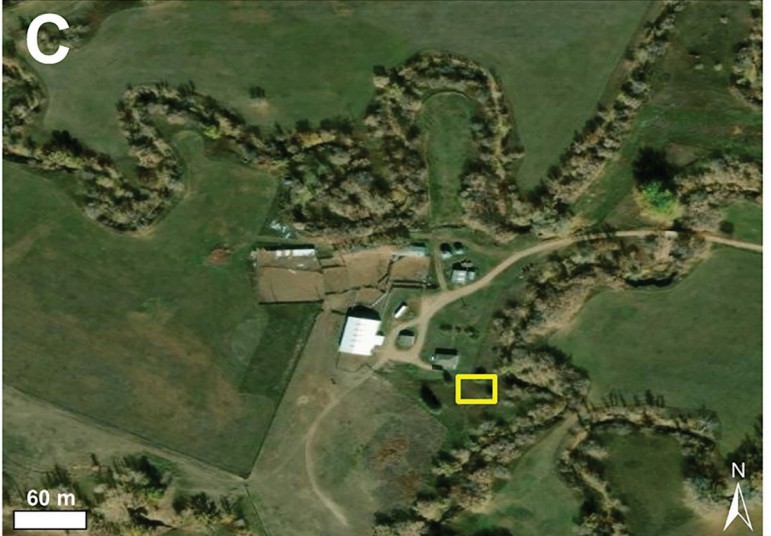

**Fig 2. Geographic region and site location.** (A) South Dakota is shaded in dark grey, and the field location is indicated by the star. (B) View of the field area along the eastern margin of the Black Hills indicated by the star. (C) The site is located adjacent to Spring Creek and outlined with an orange box. Maps were produced using ESRI ArcGIS Desktop 10.7.1 using the World Imagery service layer (Source; ESRI, DigitalGlobe, GeoEye, Earthstar Geographics, CNES/Airbus DS, USDA, USGS, AeroGrid, IGN, and the GIS User Community).

towards the south until the burial or bone layer was encountered (Fig 3). Once the depth and location of the bone layer were determined, soil samples were collected from the exposed profile walls using small spatulas sterilized with 70% ethanol between sampling locations. Soils were collected at 0–5 cm (surface), 10–15, 20–25, 30–35, 40–45 (bone-bearing layer), and 45–50 cm (below the bone-bearing layer) below the present-day surface (S1 File). All soils were placed in sterile plastic bags and stored on ice during transport to the lab (<2 hr). On or within-mound samples were collected on 10 November 2021. Surface controls (0–5 cm) were collected at the same time from three locations approximately 5 m from the burial site. Controls at depth (10–15, 20–25, 30–35, and 40–45 cm) were collected on 17 November 2021 from the same control locations previously sampled at the surface. Samples were collected the following week due to increasing wind speeds (gusts of 20–30 mph) and time constraints. The weather from 10 to 17 November was abnormally warm (-2.5 to 11.9°C; 4.8°C average) and dry (0.05 cm rain) for November in the area (weather station USW00024090).

The rest of the burial site was left undisturbed for future study. The trench was left exposed and not infilled. Exposed bones, including the skull (Fig 3C), vertebrae, and forelimbs were collected but are not examined here. All other bones and skeletal remains were left in situ. Adhering soil and blowfly puparia were recovered and examined for stable carbon and nitrogen isotopes, described below.

## Soil processing and analyses

Soil samples were homogenized within 4 hr of sample collection. Soil was processed to remove any material > 2 mm, including vegetation/roots, rocks, insects, or other organic material. Soils were subsampled within 12 hr of collection for downstream analyses. Soil gravimetric moisture was measured in duplicate, allowing soils to dry for at least 48 hr at 105°C. Soil

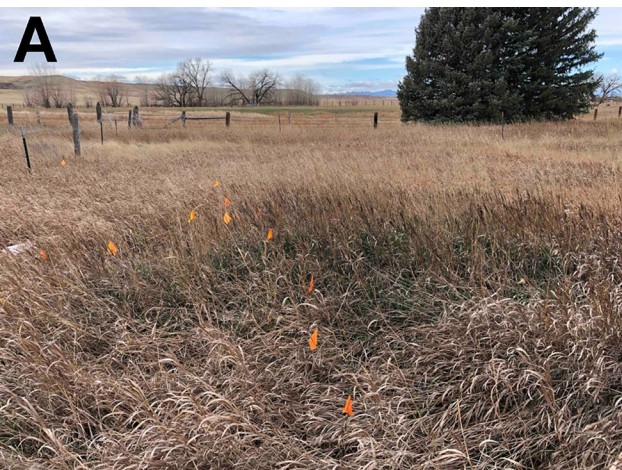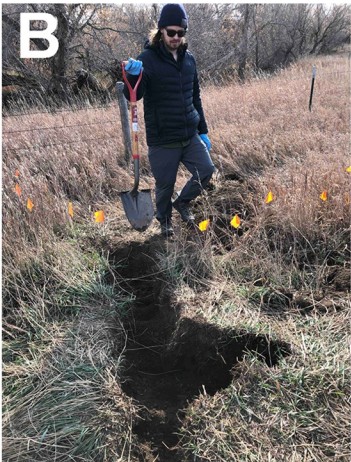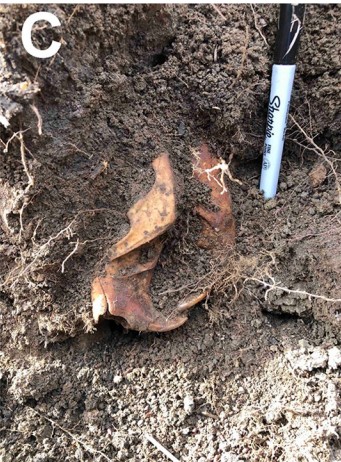

**Fig 3. Photos of the site and excavation.** (A) View of the mound facing west-southwest with orange flags (each spaced 0.5 m) marking transects across the surface of the mound (surface transect data not presented here). (B) Trench efforts to locate the buried remains along the top of the mound. (C) The bone layer and exposed beaver skull (anterior region).

gravimetric moisture was also used to present data on a dry-weight basis (i.e., respiration rates, dissolved organic carbon). Soil pH and conductivity were measured on soil slurries (10 mL of deionized water with 5 g of field moist soil) using benchtop meters and probes (Accumet, AB200) after calibration. The color of field-moist soil was assigned using a Munsell Soil chart [33].

Respiration rate, an estimate of heterotrophic bacterial activity, was measured using 6 g of soil in 60 mL clear glass serum vials. Evolved headspace $CO_2$ was measured immediately after capping ($T_0$) with chlorobutyl septa, and 24 hr after incubation in the dark at room temperature ($T_{24}$). Analyses were conducted on an infrared gas analyzer (LI-830, Licor Inc.), with calibration standards of 1000 ppm, 1%, and 5% $CO_2$ (GASCO). For each sample, $CO_2$ was measured in duplicate and reported as an average.

Field moist soils were processed for extractions using 0.5 M $K_2SO_4$ (1:4, soil:solution) using acid-washed glass jars. Soils were allowed to shake at room temperature at 150 rpm for 4 hr. After allowing extracts to settle for approximately 30 minutes, samples were vacuum filtered using a Hoefer manifold and glass microfiber filters (Ahlstrom, grade 121). Controls consisting of 0.5 M $K_2SO_4$ were processed at the same time, including shaking and filtration. Soil extracts and control samples were collected in sterile 15 mL centrifuge tubes (Falcon) and stored at –20°C until analysis. Soil extracts and controls were analyzed for dissolved organic carbon (DOC) content using a persulfate oxidation method [34]. In brief, 2 mL of soil extract were digested with 2 mL of persulfate reagent overnight at 80°C, and the $CO_2$ released was measured on an infrared gas analyzer (LI-830, Licor Inc.). Two carbon sources, potassium hydrogen phthalate or KHP and a mixture of glycine and dextrose, were used to generate calibration curves from 100 ppm C stock solutions. Standards were diluted using 0.5 M $K_2SO_4$ to account for any matrix effects. Standards included 0, 10, 50, and 100 ppm C, and samples were digested following the same protocol for soil extracts. Any soil extract digestions that fell above the 100 ppm curve were re-digested at a dilution, typically 1:1. Calculated DOC concentrations are presented on a gram dry weigh basis.

Soil elemental chemistry was evaluated using inductively coupled plasma mass spectrometry (ICP-MS) on two soil extract types: water soluble extracts and pseudototal (non-silicate bound) strong acid extracts. Water extractions were performed by combining 3.34 g of soil with 20 mL of water in a 50 mL conical centrifuge tube and shaking at 160 rpm for 4 hours [35]. After shaking samples were allowed to settle overnight and 1 mL was combined with 1 mL of 20% nitric acid and 8 mL deionized water. Acid extractions were performed using an Anton Paar Multiwave 5000 Microwave Reaction system following EPA Method 3051a. Briefly, ~500 mg of soil was combined with 9 mL of 70% nitric acid (TraceMetal Grade, Fisher Scientific) and 3 mL of 37% hydrochloric acid (TraceMetal Grade, Fisher Scientific). Samples were extracted using the built-in EPA 3051a method on the microwave extraction system. Following digestion samples were brought to a final volume of 50 mL with deionized water and allowed to settle overnight. Samples were then diluted an additional 100 times with 2% nitric acid and analyzed via ICP-MS. Acid extraction recoveries were assessed by extracting Enviro-MAT SS-1 Soil Standard (SCP Science) with each extraction batch and ensuring results were within the certified tolerance interval.

Both water and acid extracts were analyzed with an Agilent 7900 ICP-MS in the Engineering and Mining Experiment Station at South Dakota Mines. The ICP-MS was operated in the helium collision mode and quantification was performed using a multipoint external calibration. Standards were prepared by serial dilution of a certified multielement standard solution (28 SCP-AES, SCP Science). Instrument calibration was confirmed using a second certified multielement standard solution (ICP-MSCS-PE3, High Purity Standards). Internal standards (SCP-ICS7, SCP Science) were added to all samples at 1 ppm and used to correct for interferences and instrumental drift. Data processing and correction were performed in ICP-MS

Masshunter (v. 4.1, Agilent Technologies). Where measured values fell below the detection limits, ½ of the LLOQ (lower limit of quantification) were used for statistical analyses.

Approximately 500 mg of oven-dried soils generated during gravimetric moisture analyses were powdered using an agate mortar and pestle and stored in sterile 1.5 mL microcentrifuge tubes. Depending on sampling depth, 20–25 mg (surface) to 50 mg (deepest samples) of powdered soil were transferred into 5 x 9 mm tin capsules (Costech) and weighed. Additionally, 20 mg of oven-dried blowfly puparia were powdered in an agate mortar and pestle and 1 mg transferred into tin capsules. Carbon and nitrogen stable isotopic analyses were conducted at the Stable Isotope Facility at University of California, Davis on an EA-IRMS. Additional standards used to ensure quality control and assurance include IAEA-600, USGS40, USGS41, USGS42, USGS43, USGS61, USGS64, and USGS65. Values were adjusted for linearity and drift based on analyses of in-house reference materials (Nylon powder and alfalfa flour). Precision (mean of the standard deviation for reference materials) and accuracy were ±0.10‰ and ±0.04‰ for $\delta^{13}C$, and ±0.09‰ and ±0.09‰ for $\delta^{15}N$. Vienna Pee Dee Belemnite was the primary isotopic reference material for $\delta^{13}C$ measurements and air was used for $\delta^{15}N$.

## Multivariate and statistical analyses

Data were visualized using multivariate analyses including principal component analyses (PCA) using PAST [36]. PCAs provided a visualization method to determine which soil elemental data best explained observed differences between control and grave soils. Data identified as contributing significantly to separation in multivariate space were analyzed statistically. Statistical analyses were conducted in R studio (v.4.2.2) [37]. Data were analyzed to determine if there were significant trends as a function of depth and treatment using non-additive two-way ANOVAs (treatment + depth). A non-additive approach was selected because the effects of one factor (treatment) were possibly different at all levels of the other factor (depth). In other words, the effects of the burial were not expected to equally impact the soil profiles at all depths. Post-hoc Tukey HSD testing was used to determine significant ($p < 0.05$) differences between depths. To test for significance in the interaction between treatment and depth (treatment*depth), additional two-way ANOVAs were conducted.

## Results

### Excavation and bone recovery

The bone layer was approximately 40–45 cm below the surface of the mound. Soil color was uniform and ranged from dark yellowish brown (10YR 4/4) to brown (10YR 4/3) (Fig 4; S1 Table). Control soils collected from three locations at least 5 m from the mound consisted of soils that were typically very dark brown (10YR 2/2) to dark brown (10YR 3/3) and dark yellowish brown (10YR 3/4) at depth. The soil recovered from the bone layer as well as above (30–35 cm) and below (45–50 cm) were similar colors and tended to contain more yellow hues than control soils at similar depths.

There were abundant blowfly puparia and roots associated with the recovered beaver skull and bones, particularly the ventral surface of the skull (Fig 5). The bones were orange to brown in color, demonstrating discoloration compared to fresh bone. There were no soft tissues remaining on the bones recovered.

### Soil physiochemistry

Soil pH in the grave profiles was 8.56 ± 0.17 at the surface, and gradually declined to 8.14 ± 0.04 at 30–35 cm depth (Fig 4; Table 2). Soil pH significantly dropped to 6.39 ± 1.03 within the bone layer and to 5.19 ± 0.21 below the bone layer compared to soils above ($p = 0.0019$, f = 0.734). Control soils were similar to grave soils until 40–45 cm depth, where

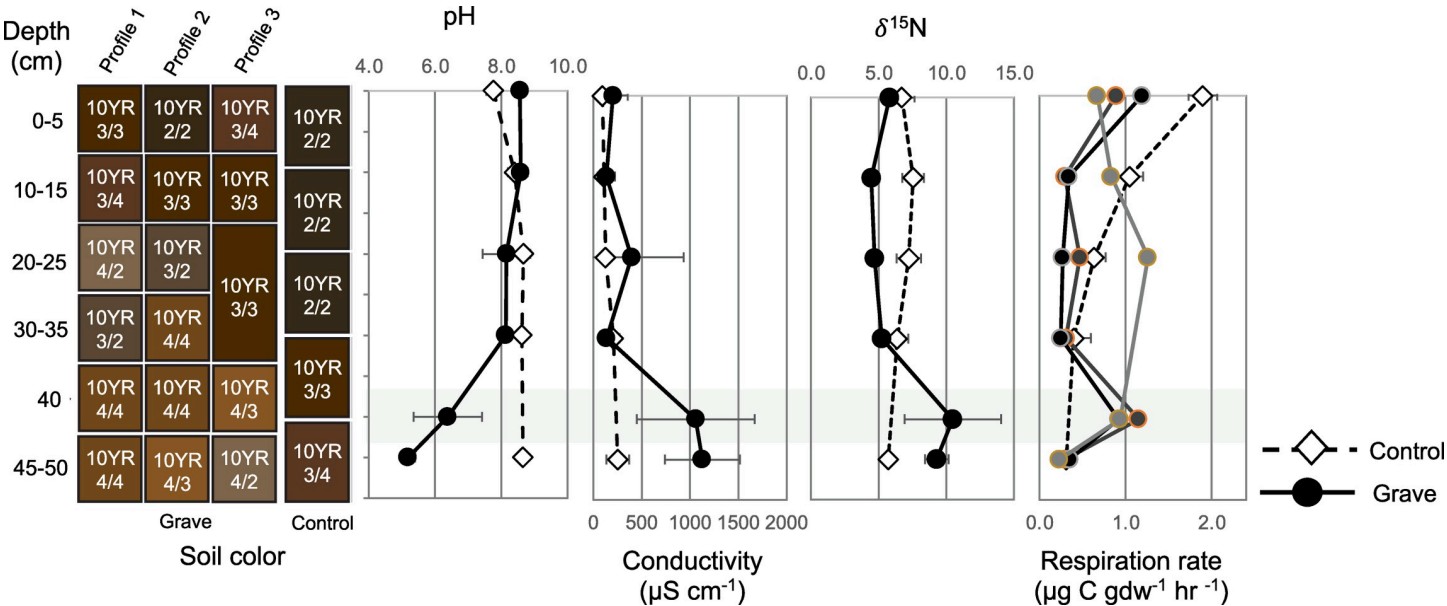

**Fig 4. Changes to soil physical and chemical properties with depth in grave and control samples.** A representative control soil profile is included for comparison. The grey horizontal bar indicates the bone layer at ~40 cm depth. The three grave profiles are displayed for respiration rate because one profile (3) exhibited elevated respiration depths above the bone-bearing layer compared to the two other profiles.

grave soils were significantly more acidic than controls (p = 0.0296, f = 5.299). At depth, control soils were pH 8.65 ± 0.15.

The conductivity of control soils remained relatively constant from the surface to depth, with values ranging from 93.33 ± 56.16 at the surface to 250.9 ± 117.6 μS cm$^{-1}$ at depth (Fig 4; Table 2). Grave soils exhibited a similar trend and similar conductivity until the bone layer when conductivity significantly increased to 1058 ± 609.2 μS cm$^{-1}$ (p = 0.0009, f = 6.525). Elevated conductivity persisted beneath the bone layer as well (1128 ± 386.9 μS cm$^{-1}$).

Gravimetric moisture in control soils decreased from 0.33 ± 0.01 in surface soils to 0.17 ± 0.03 at depth (Table 2). In contrast, grave soils exhibited no change in gravimetric moisture from the surface (0.25 ± 0.01) to depth (0.23 ± 0.00). Respiration rates were elevated in the bone layer compared to controls (Fig 4). Rates were similar between control and grave soils below the bone layer as well as at 30–35 and 20–25 cm.

## Multivariate analysis of elemental chemistry

Multivariate analyses (PCA) of trace and major element chemistry of soil from both water and acid extractions separate strongly in multivariate space (Fig 6A and 6B). Water extracted elemental data separation in multivariate space is largely explained by differences in Fe and Al, with lesser influence from barium (Ba), titanium (Ti), and Na (principal component 1 (PC1) accounts for 66.97% of variability) (Fig 6A). PC2 (15.65%) was positively influenced by K, with lesser contributions from manganese (Mn), Na, Mg, and Ca. Acid extracts separate strongly along PC 1 (63.37% of variability) and is driven by Na and many trace metals, including Fe. PC 2 (20.09% of variability) is strongly separated by P and Ca as well as Mg.

## Water extractable soil major and trace elemental chemistry

Water soluble elemental concentrations in the soils varied between depths and between grave and control soils (Fig 7; S1 File). Ca, magnesium (Mg), strontium (Sr), and sodium (Na)

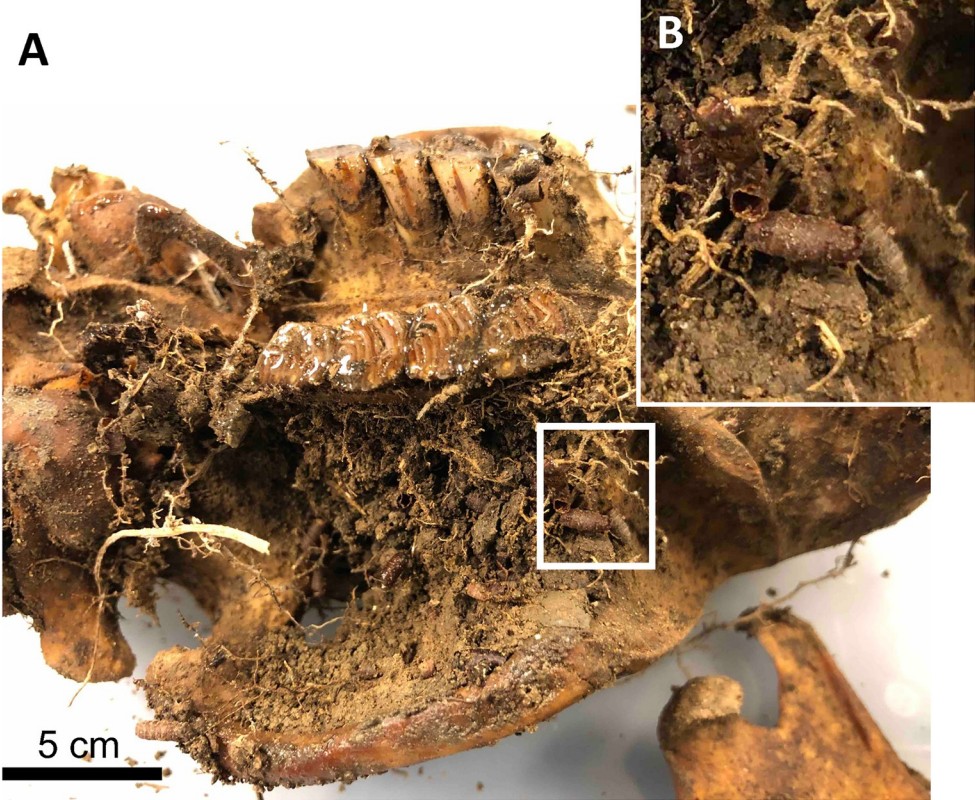

**Fig 5. Oblique, right-lateral and ventral view of the recovered beaver skull.** (A) Soil adhered to the bone surface and infilled the cranial cavity. (B) Enlarged view of adhering soil with abundant roots and blowfly puparia (inset image scale is approximately 5 cm).

concentrations were greater at the bone layer compared to soils above (0–5, 10–15, 20–25, and 30–35 cm). While Ca concentrations significantly increased with depth, there was no significant difference between control and grave profiles (p = 0.156, f = 2.135) (Fig 7). Although the grave soils trend towards greater Ca concentrations at depth, because there was high variability between replicates, this was not significant. Aluminum (Al), iron (Fe), and potassium (K) concentrations were significantly lower in the bone layer compared to the surface (0–5 cm). There was an overall trend of higher Al concentrations in control vs. grave soils, but this trend was not significant (p = 0.286, f = 1.187).

Mg, Na, Sr, and K exhibited similar significant differences as a function of both depth and treatment (Fig 8; S2 Table). Differences were driven by Mg enrichment, which increased with depth in grave soil. Control soil Mg concentrations did not increase with depth and were significantly different from grave soils (p = 0.0002, f = 8.428). Na likewise displayed significant differences with increasing depth (all p <0.009; S1 File) and between treatments (p = 0.002, f = 11.98). Grave soil Na increased with depth, resulting in significant differences between grave and below bone layer soils with all soil above. Control soils exhibit some increase in Na concentrations with depth, but this is not significant. Sr and K both were significantly different between grave and control soils (p = 0.002, f value = 11.27; p = 0.0192, f = 6.237) and with depth. Sr concentrations in grave soil exhibited a very similar trend to elements like Al, which increased with depth (S2 Table). K concentrations, however, were greatest at the surface and decreased with depth. A similar pattern was also observed in control soils.

**Table 2. Geochemical and physiochemical data for grave and control soils.** Values are based on means of duplicate sampling (except for 30–35 cm grave sample, which is based on duplicate samples), and SD indicates standard deviations.

| | Depth interval (cm) | Conductivity (µS cm⁻¹) | SD | pH | SD | Gravimetric Moisture | SD | Respiration (µg C gdw⁻¹ hr⁻¹) | SD | DOC (mg gdw⁻¹) | SD | C:N | SD | N (%) | SD | C (%) | SD | $\delta^{13}C_{VPDB}$ (‰) | SD | $\delta^{15}N_{Air}$ (‰) | SD |
|---|---|---|---|---|---|---|---|---|---|---|---|---|---|---|---|---|---|---|---|---|---|
| Grave | 0–5 | 204.2 | 150.2 | 8.56 | 0.17 | 0.25 | 0.01 | 0.92 | 0.261 | 16.74 | 2.05 | 12.2 | 1.0 | 0.289 | 0.102 | 3.45 | 0.97 | -19.6 | 3.23 | 5.79 | 0.37 |
| | 10–15 | 134.7 | 89.6 | 8.59 | 0.21 | 0.26 | 0.04 | 0.493 | 0.297 | 15.21 | 6.19 | 13.9 | 2.3 | 0.204 | 0.108 | 2.67 | 0.91 | -16.87 | 4.78 | 4.46 | 0.6 |
| | 20–25 | 395.7 | 539.1 | 8.16 | 0.72 | 0.24 | 0.01 | 0.668 | 0.524 | 20.95 | 9.68 | 13.4 | 1.4 | 0.179 | 0.056 | 2.35 | 0.49 | -16.41 | 3.75 | 4.71 | 0.15 |
| | 30–35 | 137.5 | 2.97 | 8.14 | 0.04 | 0.26 | 0.01 | 0.279 | 0.043 | 17 | 2.01 | 10.3 | 0.6 | 0.186 | 0.033 | 1.9 | 0.23 | -18.78 | 2.6 | 5.25 | 0.02 |
| | bone layer (~40 cm) | 1058 | 609.2 | 6.39 | 1.03 | 0.24 | 0.02 | 1.004 | 0.13 | 40.3 | 2.78 | 9 | 0.2 | 0.503 | 0.301 | 4.52 | 2.74 | -23.6 | 3.15 | 10.48 | 3.57 |
| | below bone layer (~45 cm) | 1128 | 386.9 | 5.19 | 0.21 | 0.23 | 0.00 | 0.295 | 0.058 | 16.94 | 1.4 | 7.8 | 0.5 | 0.409 | 0.021 | 3.21 | 0.23 | -24.1 | 0.2 | 9.3 | 0.87 |
| Control | 0–5 | 93.33 | 56.16 | 7.76 | 0.21 | 0.33 | 0.01 | 1.897 | 0.166 | 25.46 | 3.17 | 11.8 | 0.8 | 0.599 | 0.17 | 6.98 | 1.54 | -25.11 | 1.23 | 6.28 | 0.55 |
| | 10–15 | 110.1 | 57.89 | 8.4 | 0.2 | 0.23 | 0.00 | 1.049 | 0.154 | 11.5 | 0.99 | 11.7 | 0.9 | 0.349 | 0.085 | 4.02 | 0.65 | -20.9 | 1.81 | 7.55 | 0.79 |
| | 20–25 | 124.5 | 46.86 | 8.67 | 0.16 | 0.2 | 0.03 | 0.636 | 0.135 | 7.33 | 4.75 | 14.6 | 1.1 | 0.207 | 0.019 | 3 | 9.6 | -16.61 | 1.04 | 7.23 | 0.9 |
| | 30–35 | 205.3 | 64.85 | 8.62 | 0.06 | 0.18 | 0.04 | 0.399 | 0.199 | 8.22 | 1.4 | 20.6 | 2.3 | 0.14 | 0.012 | 2.87 | 0.28 | -13.28 | 0.72 | 6.37 | 0.81 |
| | 40–45 | 250.9 | 117.6 | 8.65 | 0.15 | 0.17 | 0.03 | 0.313 | 0.019 | 9.21 | 2.0 | 23.6 | 1.0 | 0.108 | 0.018 | 2.75 | 0.33 | -12.27 | 0.57 | 5.71 | 0.42 |

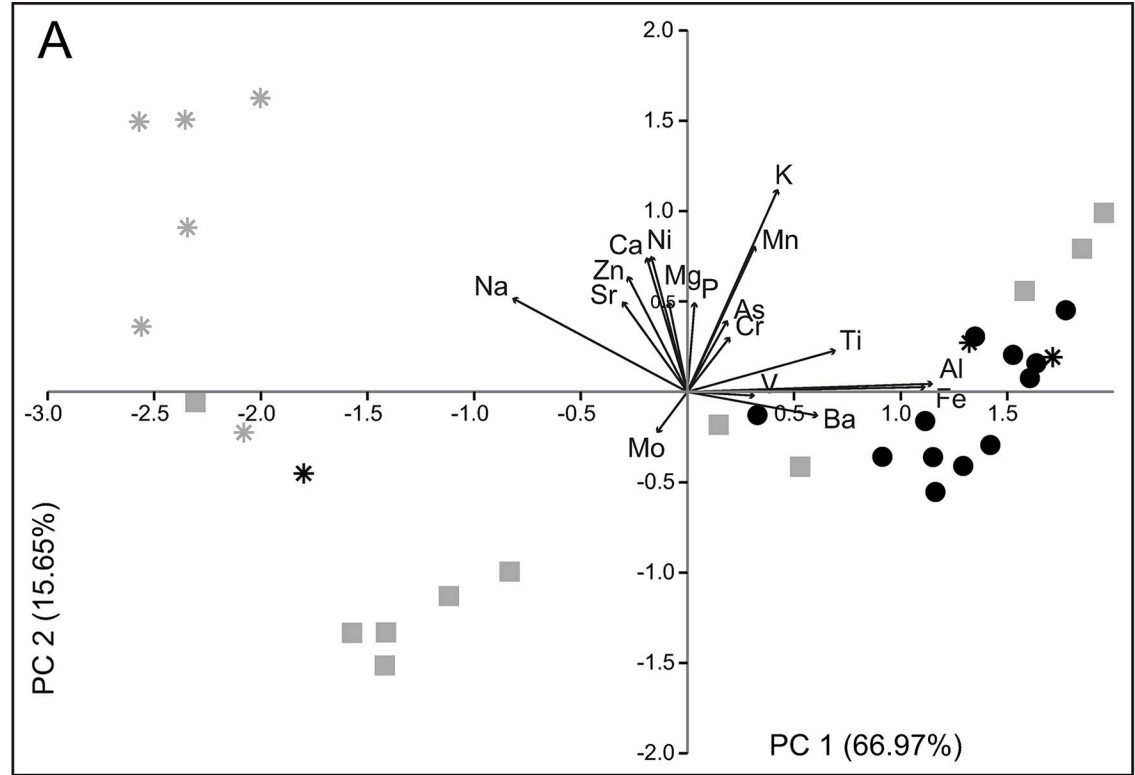

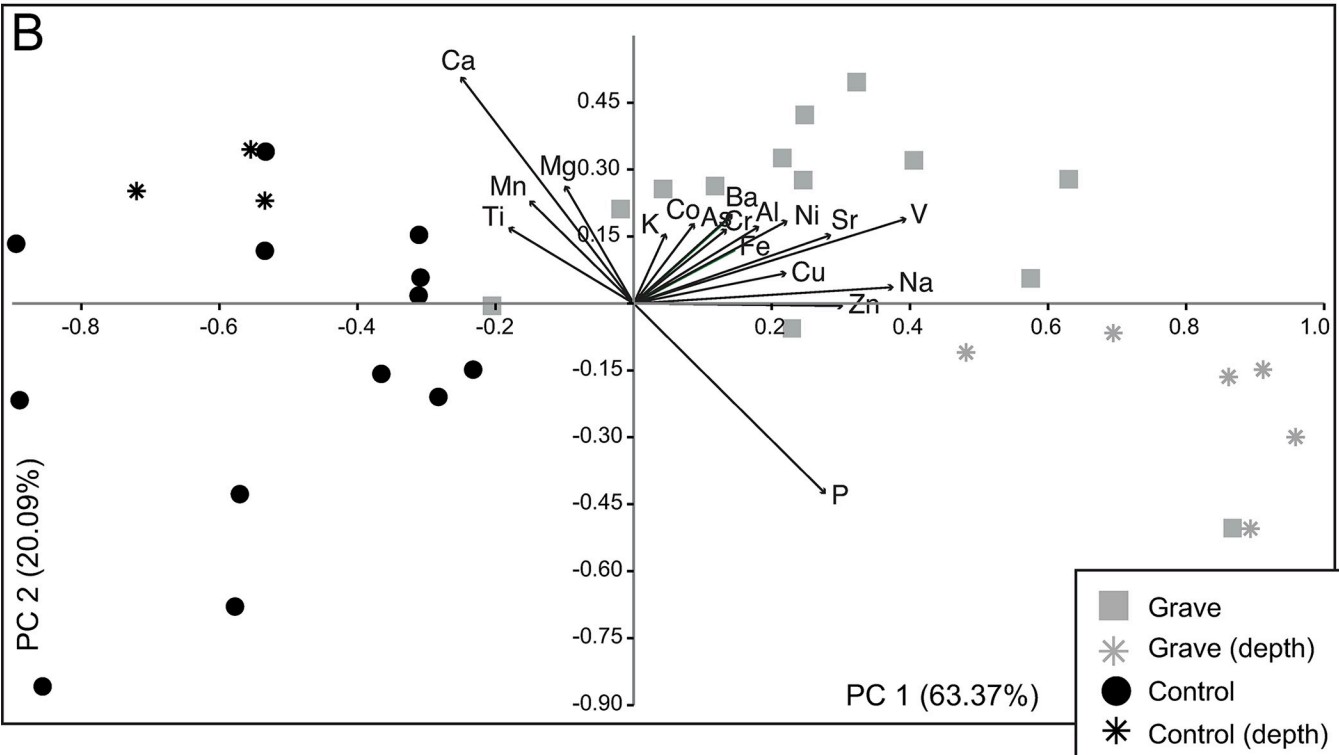

**Fig 6. Principal components analysis of log normalized major and trace element chemistry of grave and control soils.** (A) Water extract and (B) acid extract samples are differentiated based on treatment (control and grave). Samples from depth (bone layer and below) are denoted by star symbols.

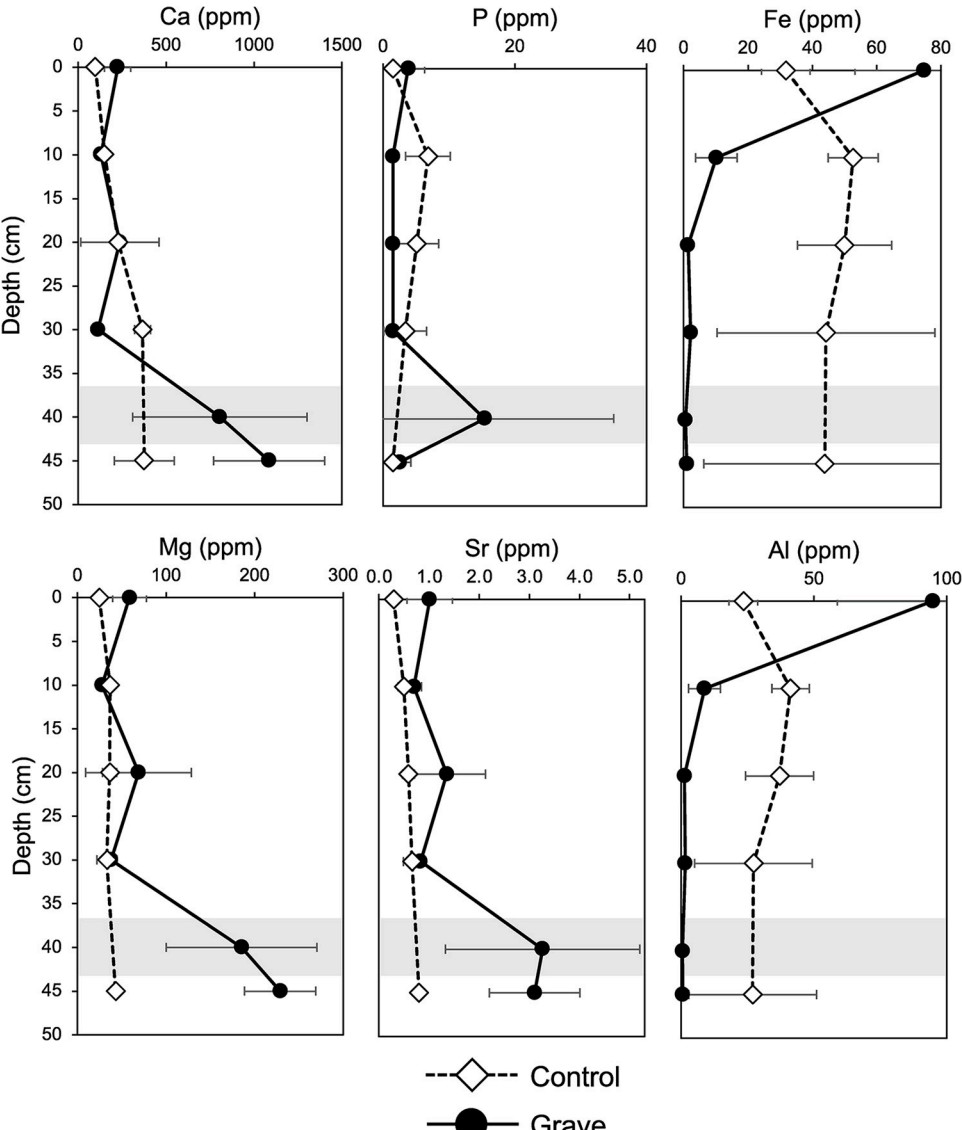

**Fig 7. Concentrations of selected elements in control and grave soils from water extractions.** Data are presented as a function of depth below the surface. The interval highlighted in grey indicates the approximate location of the bone layer in grave soils.

In contrast to significant differences with depth or between treatments observed for many major and trace elements, P exhibited no significant changes with depth (all $p > 0.05$) or between grave and control samples ($p = 0.889$, $f = 0.020$), and the interactive effects of treatment*depth were also not significant ($p = 0.468$, $f = 0.924$). Soils from the bone layer were highly variable (15.43 ppm ± 19.6% relative standard deviation), and half of the samples from the bone layer from the grave contained P concentrations below detection. While there was a trend of increasing P in the bone layer, statistically, this was not significant. Fe concentrations from water extractions also did not exhibit a significant difference by depth ($p = 0.9937$). However, there was a significant difference between control and grave soils, driven by higher concentrations of Fe in control soils ($p = 0.005$; $f = 9.23$).

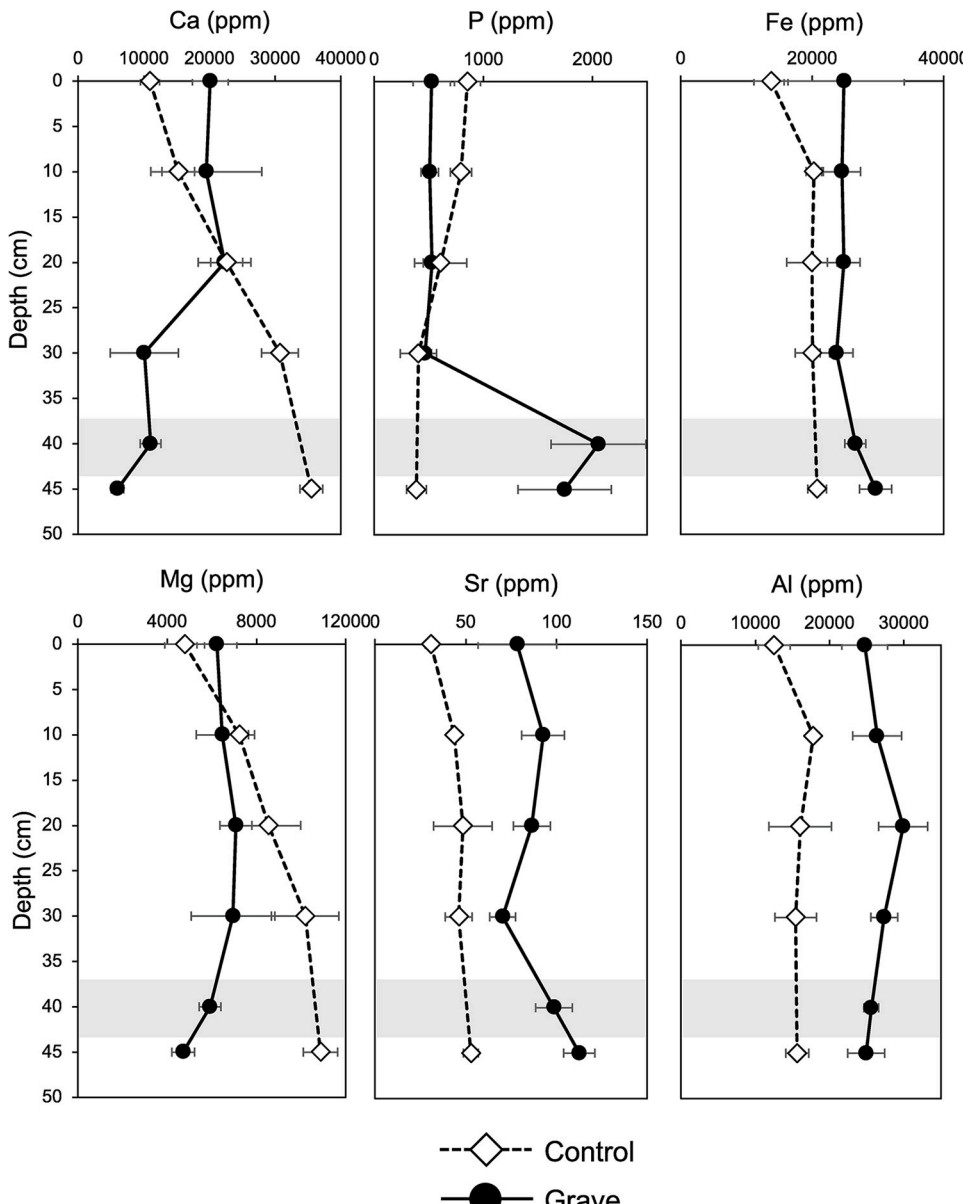

**Fig 8. Concentrations of selected elements in control and grave soils from acid extractions.** Data are presented as a function of depth below the surface. The interval highlighted in grey indicates the approximate location of the bone-bearing layer.

## Acid extractable soil major and trace elemental chemistry

The strong acid extractable elemental concentrations also varied with depth and between the control and grave soils (Fig 8; S1 File). Some elements displayed similar trends with respect to differences between soils at the surface and at depth, differences between control and grave soils, typically with greater concentrations of trace elements at the bone layer. For example, Sr concentrations were different between treatments ($p < 0.00001$, $f = 1111.93$) with the highest concentrations observed in grave soils at the bone layer (Fig 8), a trend mirrored in water extracted elemental chemistry as well (Fig 7). Ca concentrations from acid extractions were significantly different between control and grave soils ($p = 0.0225$, $f = 5.812$), driven by

decreased extractable Ca in grave soils at the bone layer. This contrast with water extractions, where Ca increased at the bone layer.

P significantly increased with depth in grave soils with the greatest concentrations observed in and below the bone layer (p = 0.0009, f = 6.289). However, there was no statistically significant difference between treatments (p = 0.129). Acid extractable Mg concentrations were significantly different with depth (p = 0.026, f = 3.222) and between treatments (p = 0.0014, f = 12.439). Mg concentrations decreased with depth in grave soils and increased in control samples (Fig 8).

In contrast to results observed with water extracts, Fe concentrations were significantly greater in grave soil compared to controls (p < 0.00001, f = 22.821) and there were significant increases in Fe with depth in both the grave and control soils (S3 Table). Al also exhibited a similar trend (and opposite to results of water extracts), with significantly more Al in grave soils compared to controls (p < 0.00001, f = 150.160). There were no significant differences in Al with depth.

### Soil carbon and nitrogen pools

The $\delta^{15}N$ composition of control soils remained relatively constant with depth, ranging from 7.55 ± 0.79‰ (10 cm depth) to 5.71 ± 0.42‰ (40 cm depth) (Fig 4; S3 Table). In contrast, the grave soils ranged from 5.79 ± 0.37 (surface) and 4.46 ± 0.60‰ (10 cm depth), before significantly increasing to 10.48 ± 3.57‰ at the bone layer (p = 0.0479, f = 2.778) (Fig 4).

The $\delta^{13}C$ composition of control soils became more enriched with increasing depth, although this was not statistically significant (p = 0.0783). Soils carbon stable isotopic composition was –25.11 ± 1.23‰ at the surface and increased to –12.27 ± 0.57‰ at depth. Grave soil carbon stable isotopic composition exhibited the opposite trend, becoming more depleted with depth (surface: –19.60 ± 3.23‰; depth: –24.10 ± 0.20‰) (Table 2).

The carbon to nitrogen ratios (C:N) of control soils increased from 11.8 at the surface to 23.6 at depth (Table 2). This is driven by both a decrease in C% as well as a decrease in N%, which is approximately 6 times lower in soils at depth compared to the surface. Grave soils displayed a decrease in C:N ratio with depth. Surface soils were similar to controls (C:N = 12.2), but differed significantly at depth (C:N = 7.8) (p = 0.0012, f = 13.21). N% in soils at the bone layer was almost double the nitrogen content observed at the surface of the grave.

### Discussion

Despite over two decades of decomposition, buried carrion continued to exert a significant influence on soil biogeochemistry. Several key parameters in soil that are indicative of recent impacts from carrion decay in surface systems, including pH, conductivity, and $\delta^{15}N$, are still perturbed in the burial compared to control soils from equivalent depths. In surface decay settings, soils can either become more acidic or more alkaline compared to controls, with a variety of factors controlling pH changes, and pH can also change through time [24]. Here, observed soil acidity may be a reflection of enhanced organic acid release from soil microorganisms targeting P or potentially collagen in bone. Additionally (or alternatively), the observed increase in respiration rates within the bone layer soils suggests that heterotrophic microorganisms are actively producing $CO_2$, potentially through the breakdown of residual organic carbon from carcass-derived soft tissues or potentially bone-derived collagen, both of which could be reflected in the elevated soil DOC. As a consequence of enhanced respiration rates, soil pH would then decrease through the dissociation of $CO_2$ into water to from carbonic acid ($H_2CO_3$) and a proton ($H^+$) [38].

In addition to potential impacts to (and from) soil microbial communities, the observed acidity at the bone layer could help to explain the observed increase in P concentrations, and

to a lesser extent, the trends in Ca concentrations. Bone, which consists primarily of bioapatite, a Ca and P-bearing mineral, exhibits increased solubility under acidic conditions [39]. In a study modeling the thermodynamic stability of apatite minerals similar to bioapatite in a shallow burial in a wetland setting, even under slightly acidic conditions (i.e., pH ~6.50), bone is expected to start to dissolve [40]. The results observed here with respect to soil biogeochemistry suggest that the pH and ultimately bone preservation or dissolution is likely controlled by: (1) C pools and cycling, (2) oxygen availability, (3) microbial community composition and metabolic activity, particularly heterotrophs, and (4) water availability, which is related to both climate and soil type. While the microbial community membership was not explored in the present study, further work focusing on the metabolic capabilities of microbial taxa present may help to resolve if genes related to phosphorus solubilization are active in grave soils.

Soil element (major, trace, and minor) data suggest that the presence of carcasses, and ultimately bones and teeth, continue to impact soil chemistry long after burial and soft tissue decomposition. Elements in soil are distributed in a variety of phases and occur in many chemical forms that exhibit a range of mobility and bioavailability [41, 42]. This study assessed both the water soluble (or extractable) fraction comprised of free ions or ions complexed with soluble organic matter or ligands, and the strong acid extractable (or pseudototal) fraction associated with most soil components except for silicates and other highly recalcitrant phases [43]. Accordingly, the water extractable fraction represents the most mobile and (bio)available elements, while the strong acid extractable fraction represents the elements that may be mobilized with a change in environmental conditions [44]. Changes in the concentrations of elements in each of these pools may provide insight into exchanges occurring between the bones and soil phases, and between the bones and biological communities present. However, study of the effects of animal decomposition on the elemental composition of soils in both burial and surface settings is limited.

The mineral component of bone consists of hydroxylapatite ($Ca_{10}(PO_4)_6(OH)_2$), with a range of other ions present including carbonate, Na, and Mg. Because Ca and P are often limiting in soil environments, the concentrations of Ca and P were expected to be perturbed in burial soils due to the presence and dissolution of bone. Additionally, a prior archaeological study investigating soils from a ~4,500 year old grave site in the Czech Republic found elevated Ca, P, and other trace elements [45]. In their study, Asare et al. [45] observed elevated P concentrations in the strong acid extractable fraction of grave soils compared to control locations, suggesting that subsurface decay and the presence of bones have the potential to impact ecosystems for thousands of years [45]. In a prior study by Keenan and Engel [46], after 3 years of burial in a wetland, alligator bones physically and chemically were altered, including changes to calcium in the bone. Their results demonstrate that exchange between soils and bones can occur relatively rapidly after soft tissue degradation [46]. Decomposing animals and their bones contribute to soil chemistry very early during decay after soft tissue removal and elemental exchange with soil may persist over long periods of time.

In the present study, acid-extractable Ca concentrations at and below the bone-bearing layer were significantly lower in grave soils, while a slight non-significant increase in Ca was found in the water extractable fraction. The decreased Ca concentrations associated with burial soils contrasts with previous results and may reflect soil pH, which was acidic. Increased acidity would favor mobilization of Ca as other metals like Fe and Al preferentially bind to P under acidic conditions. Conversely, significantly increased P concentrations were observed in the acid-extractable fraction at and below the bone-bearing layer in grave soils with no significant difference in the water-extractable fraction. Once P is mobilized from bone into soil it can persist as a complex or bound with Ca, Al, or Fe, depending on soil pH [47]. The results here, notably the elevated P concentrations in the acid extractable pool at the bone-bearing layer

compared to the water extractable pool suggests that under increased acidity, P is more mobile. Observed acidity at this layer suggests that P may be bioavailable in the bone-bearing layer with release driven directly (i.e., enzymatic) or indirectly (i.e., $CO_2$ release) by biological activity. The concentration of other trace elements (e.g., Fe, Mg, Sr, Al) were also variable between grave versus control, depth, and acid versus water extractable fraction likely reflecting the interplay of biological, chemical, and physical processes due to decomposition. Sequential extractions of these elements could help to resolve how these elements are bound and what physiochemical processes may be mobilizing and/or sequestering them. Regardless of the cause or causes, these results indicate that vertebrate decomposition in burial settings has significant effects on the elemental composition of soils over decadal timescales.

In addition to long-term disruptions to C pools and cycling, N remained altered in soil analyzed from the bone layer. Significant $\delta^{15}$N stable isotopic enrichment, elevated N concentrations (as %N), and perturbed C:N ratios suggest that N pools and availability are still disrupted compared to control soils. Similar observations from surface decay settings were on timescales of years [16], with some systems returning to background conditions after 5 years [4]. The long-lasting disruption to N in the subsurface may serve as a marker of decomposition that could provide a method of estimating time since placement and decomposition. Here, assuming that decomposing beavers in the subsurface result in similar isotopic enrichment to that observed during surface decay (i.e., 8 to 9‰ enriched above soil background), the predicted maximum isotopic enrichment during soft tissue decay would have been approximately 13 to 14‰ about 20 years ago. With a linear rate of decline of 0.2‰ per year, the grave soil here would retain a signature of decay for an additional 30 years. By developing a better understanding of the many interacting variables such as carcass type (i.e., herbivore, carnivore, omnivore), size, climate, soil type, etc. it may be possible to utilize nitrogen stable isotopes as a proxy for postmortem interval.

At present, nutrients input from carrion are not included in terrestrial ecosystem models (i.e., carbon and nitrogen cycling and budgets) [1]. While burial settings represent a small part of overall carrion decay, the potential for these systems to result in protracted and potentially important sources of limiting nutrients such as Ca and P may result in a proportionally greater impact to systems than previously recognized. The potential for carrion to provide important insights into food web dynamics in ecosystems [48] must also integrate temporal dynamics, which are likely to vary based on decay location (i.e., surface or subsurface) as well as climate and geographic region. Bones are known to serve as resources and substrates for plants and microorganisms [49], and the persistence of bones as the final nutrient source from carrion has the potential to disproportionally impact ecosystems on spatial and temporal scales beyond traditional ecosystem models.

Further research evaluating the long-term stability of decay products and bone in burial settings across broad geographic and climatic regions may help to constrain the time and longevity of decay in terrestrial ecosystems. For forensic sciences, understanding the changes that occur to soil chemistry, including trace metals and stable isotopes, may become a useful tool for establishing postmortem intervals. From a paleontological perspective, the dynamic chemical and biological changes that occur after soft tissues are decomposed likely set the stage for dictating long-term preservation of bone over geologic time. The soil chemical environment directly influences bones and likely represents an underexplored aspect of the fossilization process.

## Conclusions

The results of this study expand our understanding of the temporal scale on which vertebrate decay influences a shallow burial in a terrestrial environment, particularly soil

biogeochemistry. Acidic soil pH at the bone bearing layer combined with elevated microbial respiration rates and DOC suggests that decomposition is still active and perturbing soil biogeochemistry, particularly C cycling. Elevated N content and enriched $\delta^{15}$N stable isotopic composition at depth also indicates long-term impacts to nitrogen cycling. Trace and major elemental chemistry of soil indicates continued inputs from carrion, specifically bones, long after soft tissues are degraded, and may serve as a critical resource for limiting nutrients in terrestrial ecosystems.

## Supporting information

**S1 File. Supplemental figures and summary of statistical tests.**
(DOCX)

**S1 Table. Sample details, soil physiochemistry, and color.**
(XLSX)

**S2 Table. Soil water extract complete dataset.**
(XLSX)

**S3 Table. Soil acid extract complete dataset.**
(XLSX)

**S4 Table. Complete stable isotope dataset.**
(XLSX)

## Acknowledgments

Kenny Brown provided land for burials, information on site location and history, and continued site access. Stacy Taylor provided the method for water-based soil extractions and discussions on method development. Dr. Philip Barton and an anonymous reviewer provided helpful comments and suggestions that improved the quality of this manuscript.

## Author Contributions

**Conceptualization:** Sarah W. Keenan, Scott R. Beeler.

**Data curation:** Sarah W. Keenan, Scott R. Beeler.

**Formal analysis:** Sarah W. Keenan, Scott R. Beeler.

**Funding acquisition:** Sarah W. Keenan.

**Investigation:** Sarah W. Keenan, Scott R. Beeler.

**Methodology:** Sarah W. Keenan, Scott R. Beeler.

**Project administration:** Sarah W. Keenan.

**Resources:** Sarah W. Keenan.

**Supervision:** Sarah W. Keenan.

**Visualization:** Sarah W. Keenan, Scott R. Beeler.

**Writing – original draft:** Sarah W. Keenan, Scott R. Beeler.

**Writing – review & editing:** Sarah W. Keenan, Scott R. Beeler.

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
