## [Decision Letter · Decision Letter 0]

21 Aug 2023

PONE-D-23-21061Long-term effects of buried vertebrate carcasses on soil biogeochemistry in the Northern Great PlainsPLOS ONE

Dear Dr. Keenan,

Thank you for submitting your manuscript to PLOS ONE. After careful consideration, we feel that it has merit but does not fully meet PLOS ONE’s publication criteria as it currently stands. Therefore, we invite you to submit a revised version of the manuscript that addresses the points raised during the review process. Both reviews are positive. Reviewer 1 suggests that more could be done with the microbial data. Reviewer 2 provides a list of needed changes and corrections. Please address each of the comments while making your revisions. If you are unable to provide additional results from the microbial data, please explain why in the manuscript.

We look forward to receiving your revised manuscript.

Kind regards,

John P. Hart, Ph.D.

Academic Editor

PLOS ONE

Journal Requirements:

5. We note that Figure 2 in your submission contain [map/satellite] images which may be copyrighted. All PLOS content is published under the Creative Commons Attribution License (CC BY 4.0), which means that the manuscript, images, and Supporting Information files will be freely available online, and any third party is permitted to access, download, copy, distribute, and use these materials in any way, even commercially, with proper attribution. For these reasons, we cannot publish previously copyrighted maps or satellite images created using proprietary data, such as Google software (Google Maps, Street View, and Earth). For more information, see our copyright guidelines: http://journals.plos.org/plosone/s/licenses-and-copyright.

Reviewers' comments:

Reviewer's Responses to Questions

**Comments to the Author**

1. Is the manuscript technically sound, and do the data support the conclusions?

Reviewer #1: Yes

Reviewer #2: Yes

2. Has the statistical analysis been performed appropriately and rigorously? 

Reviewer #1: Yes

Reviewer #2: Yes

3. Have the authors made all data underlying the findings in their manuscript fully available?

Reviewer #1: Yes

Reviewer #2: Yes

4. Is the manuscript presented in an intelligible fashion and written in standard English?

Reviewer #1: Yes

Reviewer #2: Yes

5. Review Comments to the Author

Reviewer #1: The authors present a very well written study that explores the impacts of vertebrate decomposition over a temporal scale not commonly published.

The results relating to isotopic changes over time were particularly interesting, though overall I thought that the microbial changes could have been better investigated.

Reviewer #2: Review of “Long term effects of buried vertebrate carcasses on soil biogeochemistry in the northern great plains”.

Summary:

This manuscript describes a study on soil changes associated with the decomposition of vertebrate remains 21 years after burial. A range of physicochemical measures were made at different depths above and below the remains, and compared with a nearby control site at equivalent depths. Key findings relate to pH, phosphorus, and C/N isotopes, and some interpretation of these findings are provided in the discussion.

Overall review:

I found this paper to be very well written and easy to read. It covers a fascinating topic (heterotrophic decomposition) and addresses a neglected area of study in this particular field (soil). My opinion is that this work should be published, and it will likely be of interested to many ecologists and soil scientists worldwide. The study is largely fine as is (but see comments on interaction terms), and my comments below are suggestions for the author’s consideration only.

• LL30/31 specify that P concentrations relate to your acid extractions

• L106 whats the difference between duration and longevity?

• L265 should this be “mean ‘and’ standard deviation”?

• L270 you explain that you conducted PCA first, then your ANOVAs, but your results report ANOVAs first, then PCAs. I suggest presenting your PCA results first to be consistent with your methods.

• L275 a two-way ANOVA is fine, but this should also include the interaction term treatment*depth, yes? I cant see this reported in your results section. To me, this is an important oversight, as several of your plots show a likely significant interaction term, despite no main effect of treatment. For example, Figure 7 shows that P has a higher [] at depth for your carcass site – I would expect this to be shown in a significant interaction term but its not reported, why?

• L329 change to past tense – “this was not significant”.

• L375 as above – where is the interaction term? The difference between the treatments might not be significant, sure, but this will be dependent on the depth.

• LL480-490 very nice discussion of your P results. Interesting stuff.

• Table 1 – why these “selected” studies and not others? It seems like there is an opportunity here to provide a more comprehensive and up-to-date list. Excluding forensic studies (i.e. pigs and humans), I can think of maybe another 5 or 6 surface soil studies using a range of carcass types (rabbits, deer etc) from around the world.

• Figure 1 is excellent. A nice diagram with good predictions.

• Figs 6 + 7, see comment above regarding interaction terms. These should be reported first where significant, then main effects of treatment of depth reported second.

• Figure 8, as noted above, your PCA ordination plot should be presented before your ANOVA results to be consistent with your methods.

6. PLOS authors have the option to publish the peer review history of their article (what does this mean?). If published, this will include your full peer review and any attached files.

Reviewer #1: No

Reviewer #2: **Yes: **Philip Barton

---

## [Decision Letter · Decision Letter 1]

4 Oct 2023

Long-term effects of buried vertebrate carcasses on soil biogeochemistry in the Northern Great Plains

PONE-D-23-21061R1

Dear Dr. Keenan,

We’re pleased to inform you that your manuscript has been judged scientifically suitable for publication and will be formally accepted for publication once it meets all outstanding technical requirements.

Kind regards,

John P. Hart, Ph.D.

Academic Editor

PLOS ONE

Additional Editor Comments (optional):

Reviewers' comments:

Reviewer's Responses to Questions

**Comments to the Author**

1. If the authors have adequately addressed your comments raised in a previous round of review and you feel that this manuscript is now acceptable for publication, you may indicate that here to bypass the “Comments to the Author” section, enter your conflict of interest statement in the “Confidential to Editor” section, and submit your "Accept" recommendation.

Reviewer #2: All comments have been addressed

2. Is the manuscript technically sound, and do the data support the conclusions?

Reviewer #2: Yes

3. Has the statistical analysis been performed appropriately and rigorously? 

Reviewer #2: Yes

4. Have the authors made all data underlying the findings in their manuscript fully available?

Reviewer #2: Yes

5. Is the manuscript presented in an intelligible fashion and written in standard English?

Reviewer #2: Yes

6. Review Comments to the Author

Reviewer #2: Revisions have been made appropriately. A nice study that will add some important new information to the field.

7. PLOS authors have the option to publish the peer review history of their article (what does this mean?). If published, this will include your full peer review and any attached files.

Reviewer #2: **Yes: **Philip Barton

---

## [Editor Report · Acceptance letter]

11 Oct 2023

PONE-D-23-21061R1 

Long-term effects of buried vertebrate carcasses on soil biogeochemistry in the Northern Great Plains 

Dear Dr. Keenan:

I'm pleased to inform you that your manuscript has been deemed suitable for publication in PLOS ONE. Congratulations! Your manuscript is now with our production department. 

Kind regards, 

on behalf of

Dr. John P. Hart 

Academic Editor

PLOS ONE